# The Impact of High BMI on Pregnancy Outcomes and Complications in Women with PCOS Undergoing IVF—A Systematic Review and Meta-Analysis

**DOI:** 10.3390/jcm13061578

**Published:** 2024-03-10

**Authors:** Salih Atalah Alenezi, Raheela Khan, Saad Amer

**Affiliations:** 1Translational Medical Sciences, School of Medicine, Royal Derby Hospital Centre, University of Nottingham, Derby DE22 3DT, UK; salih.alenezi@nottingham.ac.uk (S.A.A.); raheela.khan@nottingham.ac.uk (R.K.); 2Prince Mohammed Bin Abdulaziz Medical City, Ministry of Health, Al-Jouf 14214, Saudi Arabia

**Keywords:** polycystic ovarian syndrome, obesity, IVF, assisted reproductive technology, pregnancy complications

## Abstract

(1) **Background**: Current evidence indicates that women with polycystic ovarian syndrome (PCOS) undergoing in vitro fertilization (IVF) have an increased likelihood of adverse pregnancy outcomes. The objective of this systematic review was to clarify the role of a PCOS-related high body mass index (BMI) in these unfavourable pregnancy outcomes. (2) **Methods**: A comprehensive search of electronic databases was conducted to identify studies investigating the impact of high BMI on pregnancy outcomes in women with PCOS undergoing IVF. RevMan software (v5.4) was used to calculate the odds ratio (OR) and 95% confidence interval (CI). (3) **Results**: Nineteen eligible studies (*n* = 7680) were identified, including 16 retrospective cohort studies (*n* = 6934), two prospective cohort studies (*n* = 525), and one cross-sectional study (*n* = 221). Pooled analysis showed significantly higher odds of clinical pregnancy (OR, 1.16 [95% CI, 1.04–1.29]; z = 2.73; *p* = 0.006; I^2^ = 30%) and livebirths (OR, 1.88 [95% CI, 1.56–2.27]; z = 6.54; *p* < 0.0001; I^2^ = 55%) in women with PCOS with a normal versus a high BMI. Meta-analysis showed significantly increased odds of miscarriages in women with PCOS with a high versus a normal BMI (OR, 0.76 [95% CI, 0.60–0.95]; z = 2.42; *p* = 0.02; I^2^ = 53%). Pooled analysis of three studies (*n* = 993) showed significantly higher ORs of gestational diabetes mellitus (OR 3.96 [95% CI 1.62–9.68]; z = 3.01; *p* = 0.003; I^2^ = 58%) and gestational hypertension (OR 2.16 [95% CI 1.32–3.54]; z = 3.05; *p* = 0.002; I^2^ = 68%) in women with PCOS with a high versus a normal BMI. Meta-analysis of three studies reported significantly greater odds of a caesarean section for women with PCOS with a high versus a normal BMI (OR 0.45 [95% CI 0.29–0.69]; z = 3.66; *p* = 0.0003; I^2^ = 0%). (4) **Conclusions**: The increased likelihood of adverse pregnancy outcomes observed in women with PCOS undergoing IVF seems to be attributable to a PCOS-related high BMI.

## 1. Introduction

Polycystic ovary syndrome (PCOS) is a very prevalent and intricate endocrine disorder in women which represents a major concern in public health [1]. Throughout their lifespan, women with PCOS encounter various health challenges including anxiety, depression, insulin resistance, hypertension, metabolic disorders, and obesity-related chronic inflammation [2,3]. Moreover, PCOS emerges as a significant contributor to infertility among women in their reproductive years due to the associated anovulation [4]. The global prevalence of PCOS in reproductive-age women is estimated to be between 8% and 13% depending on the diagnostic criteria and the demographic in focus [5,6,7,8]. The majority of diagnosed women with PCOS exhibit a high body mass index (BMI), often being categorised as overweight (BMI ≥ 25 kg/m^2^) or obese (BMI ≥ 30 kg/m^2^), with a reported prevalence of 38–88% [9]. The elevated pre-pregnancy BMI is associated with adverse pregnancy outcomes in women with PCOS, irrespective of the fertility treatment employed [10]. 

In addressing PCOS-related infertility, in vitro fertilization (IVF) remains an important and effective treatment option which is considered when ovulation induction therapies prove ineffective [11]. Although IVF is highly effective for PCOS-related infertility, IVF-conceived pregnancies are associated with an increased risk of obstetric complications compared to naturally conceived pregnancies [12]. On the other hand, according to a recent meta-analysis, PCOS is associated with a significantly higher risk of adverse pregnancy outcomes compared to women without PCOS [13]. Nevertheless, it remains uncertain as to whether these increased obstetric risks are due to PCOS, its associated high BMI, or IVF.

Although several previous studies have assessed the impact of high BMI on pregnancy outcomes among women with PCOS undergoing IVF, the results are conflicting. While some studies reported similar clinical pregnancy rates across various BMI groups among women with PCOS undergoing IVF [14,15,16], other studies reported higher pregnancy rates in lean women with PCOS compared to the high-BMI group [17,18]; meanwhile, another study revealed a higher pregnancy rate in the overweight/obese group [19]. Taking live birth as another example, one study reported no difference in live birth rates between high- and normal-BMI groups (normal BMI, 18.5–24.9 kg/m^2^) [16], while another study showed that the live birth rate is higher in normal-BMI women with PCOS undergoing IVF [18].

A recent meta-analysis published in 2022 evaluated pregnancy outcomes in women with PCOS, including miscarriage, live birth, and preterm birth rates [10]. However, this review did not assess any of the pregnancy complications such as gestational diabetes (GDM), gestational hypertension, or caesarean section rate [10]. This review also included pregnant women with PCOS who conceived via different conception methods, including ovulation induction, intrauterine insemination, and IVF [10]. Therefore, the relationship between pre-pregnancy BMI and pregnancy outcomes and complications in women with PCOS undergoing IVF remains to be determined. Furthermore, it remains uncertain as to whether the reported increased likelihood pregnancy complications among women with PCOS undergoing IVF is independent of the effects of overweight/obesity. 

We therefore undertook this systematic review and meta-analysis to clarify the effect of a high pre-pregnancy BMI on pregnancy-related outcomes and complications among women with PCOS undergoing IVF. The findings could refine individual risk evaluation for women with PCOS and inform subsequent therapeutic strategies and policies, aiming to optimise pregnancy outcomes for different BMI categories of women with PCOS planning for pregnancy.

## 2. Methods

This systematic review was executed adhering to the Preferred Reporting Items for Systematic Reviews and Meta-Analyses (PRISMA) criteria and was pre-registered on PROSPERO under the registration number CRD42023411838.

### 2.1. Eligibility Criteria for Study Selection

The search terms for this systematic literature review were formulated based on the Populations, Intervention, and Comparison, Outcome, and Study Design (PICOSD) model. The inclusion criteria were as follows: (1) the study population included women with PCOS with a high BMI; (2) the intervention was IVF; (3) the comparison group included women with PCOS with a normal BMI; (4) the outcomes encompassed clinical pregnancy, multiple pregnancy, miscarriage, live birth, preterm birth, gestational diabetes mellitus (GDM), gestational hypertension, and caesarean section; and (5) the study designs included cohort and cross-sectional studies. We excluded conference abstracts, case reports, reviews, comments, and letters. The review included only studies conducted on humans and written in English. If multiple publications from the same research team were detected, only the study with the large sample size was incorporated into the meta-analysis to prevent the duplication of cases.

### 2.2. Outcome Measures

#### 2.2.1. Main Outcomes

Clinical pregnancy rate, miscarriage rate, live birth rate, preterm birth rate, gestational hypertension, GDM, and caesarean section rate.

#### 2.2.2. Secondary Outcome Measures

Multiple pregnancy rate.

### 2.3. Search Strategy

We performed a systematic and comprehensive search of all electronic databases up to June 2023 for pertinent studies based on the above PICOSD criteria. A further manual search of the references of the selected studies was conducted to identify eligible articles. Searched databases included Medline (Ovid); EMBASE (Ovid); CENTRAL (www.thecochranelibrary.com) accessed on 26 May 2023; PubMed; Clinicaltrials.gov; the EU Clinical Trials Register; and the World Health Organization International Clinical Trials Register. A combination of the following search MeSH terms was used: “Polycystic ovary syndrome” OR “Polycystic ovaries” OR “PCOS” AND “Maternal Obesity” OR “Obesity in Pregnancy” OR “Overweight” OR “Body Mass Index” OR “BMI” AND “Miscarriage” OR “Preterm Birth” OR “Premature Births” OR “Live birth” OR “Clinical pregnancy” or “Biochemical pregnancy” OR “Multiple pregnancy” OR “Ongoing pregnancy” OR “Pregnancy-induced hypertension” OR “Gestational Hypertension” OR “Pregnancy-Induced Diabetes” OR “Gestational Diabetes Mellitus” AND “In Vitro fertilization” OR “Fertilization in Vitro” OR “IVF” OR “intracytoplasmic sperm injection” OR “ICSI”. The keywords were combined using Boolean operators for each database, as necessary. 

### 2.4. Screening and Selection of Retrieved Studies

The titles and abstracts of the sourced articles were initially assessed for their pertinence to the systematic review. Subsequently, the full texts of relevant articles were scrutinised against the specified inclusion criteria. This evaluation was performed by two separate reviewers (SAA and SA). In cases of disagreement, the two reviewers arrived at a consensus through discussion.

### 2.5. Risk of Bias Assessment

The quality of the selected studies was assessed using the Newcastle–Ottawa quality assessment scale. Three main aspects were evaluated in each study, namely selection, comparability, and outcome, with maximum scores of four, two, and three, respectively. A maximum score of 9 indicated the highest quality [20]. 

### 2.6. Data Extraction and Analysis

Data extracted from the incorporated studies encompassed details such as the primary author, publication year, study location, research design, participant attributes (including sample size, average age, and average BMI), and method of conception, infertility causes, outcome variables, and outcome definitions. 

The extracted data were uploaded into RevMan software version 5.4.1 (The Cochrane Collaboration, 2020) for meta-analysis. For dichotomous results, odds ratios (ORs) were determined with a 95% confidence interval (CI). For the meta-analysis, a random effects model was applied when heterogeneity was high. The inconsistency test (I^2^) gauged statistical heterogeneity among the selected studies. To discern any publication bias, Egger’s regression test was employed with a 2-sided *p*-value.

## 3. Results

### 3.1. Search Results

The initial search identified 7308 studies, among which 1623 were duplicate records and were removed. After screening of the remaining 5685 studies, 5638 studies were excluded as irrelevant. The full texts of the remaining 47 studies were reviewed, and 28 were excluded for the reasons detailed in Figure 1. The remaining 19 studies [14,15,16,17,18,19,21,22,23,24,25,26,27,28,29,30,31,32,33] fulfilled the eligibility criteria and were included in this review (Figure 1).

### 3.2. Study Characteristics

The review included 19 studies (*n* = 7680) investigating the effect of pre-pregnancy BMI on pregnancy outcomes in women with PCOS undergoing IVF, as summarised in Table 1. Sixteen of the nineteen studies were retrospective cohort studies (*n* = 6934), two were prospective cohort studies (*n* = 525), and one was a cross-sectional study (*n* = 221). Ten of the included studies (*n* = 6122) were performed in China. All 19 studies scored 6–8 on the Newcastle–Ottawa Scale (NOS) (Table 1). Details regarding the NOS scores are included in the Appendix A. 

Although there was a variation in the categorizations of overweight and obesity (Table 1), the cut-off used for high BMI was not significantly different between studies. Eleven studies applied the World Health Organization’s (WHO) definitions using the cut-offs of 25 kg/m^2^ to define overweight and 30 kg/m^2^ for obesity [14,15,16,18,19,21,23,25,26,31,33]. Seven studies utilised 24 kg/m^2^ and 28 kg/m^2^ as the thresholds for designating overweight and obesity, respectively [17,22,24,27,28,29,32]. Regarding the diagnostic criteria used for PCOS, the Rotterdam criteria [1] were applied in 18 studies [14,15,16,17,18,19,21,22,23,24,25,26,27,28,29,30,31,33], whereas a single study did not determine the diagnostic criteria of PCOS [32].

The study designs in the included papers comprised sixteen retrospective cohort studies [14,15,16,17,18,19,22,23,24,25,26,28,29,31,32,33], two prospective cohort studies [21,27], and one cross-sectional study [30]. All studies included women with PCOS undergoing IVF with or without ICSI. Eight studies utilised a long GnRH agonist protocol [16,19,21,22,24,27,29,30], three used a GnRH antagonist protocol [15,28,31], three used variable protocols [18,25,26], one study applied an ultra-long protocol [14], and four studies did not specify the protocol used [17,23,32,33] (Table 1). This review included only studies involving participants with PCOS with no other cause of infertility.

The included studies adopted diverse definitions for clinical pregnancy, miscarriage, live birth, gestational DM and gestational hypertension. Of the nineteen included studies, four defined clinical pregnancy as the presence of a gestational sac with a foetal heartbeat upon transvaginal ultrasonography [15,16,17,21], while one study defined it as the presence of a gestational sac [19]. Ten articles specified the timing of the ultrasound scan as between 5 and 7 weeks’ gestation [14,15,16,17,21,22,25,27,28,31]. Miscarriage was assessed in nine articles, with one defining it as pregnancy loss in the first trimester [15], three before 20 weeks’ gestation [21,23,26], and two before 28 weeks’ gestation [29,31]. The remaining three studies [14,16,17] did not specify a definition for miscarriage. Live birth was also assessed in nine studies. It was defined as an alive newborn in two studies [18,23], an alive newborn surviving more than one month after birth in one study [17], and a definition was not mentioned in the other studies [14,16,19,21,28,31]. Gestational hypertension was assessed in three studies [16,32,33], with only one study defining it as hypertension occurring for the first time during the current pregnancy after 20 weeks’ gestation, without significant proteinuria or end-organ dysfunction [33]. These three studies also assessed GDM, but neither of them provided any definition [16,32,33].

### 3.3. Clinical Pregnancy

Pooled analysis of 15 studies (*n* = 6296) demonstrated a significantly greater likelihood of clinical pregnancy among women with PCOS undergoing IVF with a normal BMI (*n* = 3722) versus those with a high BMI (*n* = 2574) (OR 1.16 [95% CI 1.04–1.29]; z = 2.67; *p* = 0.008; I^2^ = 31%). The heterogeneity between the studies was moderate (Figure 2). The results of the multiple pregnancy rates are included in the Appendix A.

### 3.4. Miscarriage

A meta-analysis of nine studies (*n* = 3015) showed a significantly higher likelihood of miscarriage among women with PCOS with a high BMI undergoing IVF versus those with a normal BMI (OR 0.69 [95% CI 0.55–0.87]; z = 3.17; *p* = 0.002; I2 = 44%). The heterogeneity between the studies was moderate (Figure 3).

### 3.5. Live Birth

Pooled data from nine studies (*n* = 2217) showed a significantly higher live birth rate among women with PCOS with a normal pre-IVF BMI compared with those with a high BMI (OR 1.98 [95% CI 1.63–2.39]; z = 7.01; *p* = < 0.00001; I^2^ = 33%). The heterogeneity between the studies was moderate (Figure 4).

### 3.6. Preterm Birth

Pooled data from six studies (*n* = 2012) found no statistical difference between women with PCOS with a high BMI undergoing IVF compared with those with a normal BMI (OR 1.20 [95% CI 0.93–1.54]; z = 1.43; *p* = 0.15; I^2^ = 28%). The heterogeneity between the studies was low (Figure 5).

### 3.7. Gestational Diabetes Mellitus (GDM)

Data from three studies (*n* = 993) revealed a statistically significant increase in the incidence of GDM for women with PCOS with a high BMI undergoing IVF versus those with a normal BMI (OR 3.96 [95% CI 1.62–9.68]; z = 3.01; *p* = 0.003; I^2^ = 58%) [16,32,33]. The heterogeneity between the studies was moderate (Figure 6).

### 3.8. Gestational Hypertension

Three studies (*n* = 993) showed a statistically significant increase in the chances of gestational hypertension for women with PCOS with a high pre-pregnancy BMI undergoing IVF versus those with a normal BMI (OR 2.16 [95% CI 1.32–3.54]; z = 3.05; *p* = 0.002; I^2^ = 68%) [16,32,33]. The heterogeneity between the studies was moderate (Figure 7).

### 3.9. Caesarean Section

Three studies showed a statistically significant increase in the chances of caesarean section for women with PCOS with a high pre-pregnancy BMI undergoing IVF compared to those with a normal BMI (OR 2.24 [95% CI 1.45–3.45]; z = 3.66; *p* = 0.0003; I^2^ = 0%) [31,32,33]. No heterogeneity was observed among the included studies (Figure 8). 

## 4. Discussion

To the best of our knowledge, this is the first meta-analysis investigating the impact of pre-pregnancy BMI on pregnancy outcomes in women with PCOS undergoing IVF. This review included 19 studies involving 7680 women with PCOS undergoing IVF. Pooled analysis of the reviewed studies showed that high pre-IVF BMI in women with PCOS was associated with lower pregnancy and live birth rates. Furthermore, a high pre-IVF BMI in women with PCOS significantly increased the risk of miscarriage, GDM, gestational hypertension and caesarean section. On the other hand, high pre-pregnancy BMI was not associated with any significant increase in the chances of preterm birth or multiple pregnancies. 

### 4.1. Comparison with Previous Studies

#### 4.1.1. Pregnancy Outcomes

The findings of this review suggest that women with PCOS who are overweight or obese have lower clinical pregnancy (OR = 0.91, *p* = 0.0003) and live birth rates (OR = 0.91, *p* = 0.01) and a higher miscarriage rate (OR = 1.24, *p* < 0.00001) compared to women with PCOS with a normal weight. In contrast, three of the cohort studies included in our review reported an elevated miscarriage rate in women with a lower BMI [17,23,26]. These findings might be attributed to bias due to a limited sample size. Two other studies showed no obvious effect of BMI on the miscarriage rates [16,29]. A previous systematic review published in 2011 by Rittenberg and co-workers reported that overweight/obese women (BMI ≥ 25) had significantly lower clinical pregnancy and live birth rates and higher miscarriage rates compared to normal-weight women [34]. However, that review was not specific for women with PCOS, and most of the included studies excluded PCOS. It is therefore possible to conclude that the adverse effect of high BMI seems to have the same adverse effect on the pregnancy outcomes of IVF regardless of the PCOS status. This is further supported by a previous cohort study by Wang et al., which revealed that the observed adverse pregnancy outcomes in women with PCOS were only attributed to high BMI, with no effect of PCOS [35]. 

The findings of this systematic review suggest that pregnancy outcomes and their complications are more likely to be associated with overweight or obesity rather than PCOS. This result aligns with a 2011 study which reported that women with a BMI ≥ 25 had significantly lower clinical pregnancy and live birth rates and a significantly higher miscarriage rate [34]. However, it contrasts with a previous systematic review that found that women with PCOS had a higher risk of adverse outcomes compared with women without PCOS [13].

Live birth rates are significantly higher in women with PCOS undergoing IVF who have a normal pre-pregnancy BMI compared to those with a high BMI. Our findings align with a prior systematic review which reported that lean pregnant women with PCOS had a significantly higher live birth rate compared to those who were overweight or obese before pregnancy [10].

Our review did not show any significant effect of a high pre-IVF BMI on the risk of preterm births in women with PCOS. These findings are consistent with a prior systematic review that reported no link between pre-pregnancy overweight/obesity in women with PCOS and preterm birth [10]. In contrast, a meta-analysis found both lower and higher maternal pregestational BMI to be linked to an increased risk of preterm birth [36]. On the other hand, another meta-analysis reported that only underweight women experienced a greater risk of preterm birth [37]. Given all these conflicting data, the impact of a high pre-IVF BMI on the chances of preterm birth in women with PCOS remains uncertain.

#### 4.1.2. GDM and Gestational Hypertension

Our data indicate that a high pre-pregnancy BMI in women with PCOS undergoing IVF is associated with an approximately 4-fold increased risk of GDM (OR 3.96 [95% CI 1.62–9.68]) and about a 2-fold increased risk of gestational hypertension (OR 2.16 [95% CI 1.32–3.54]). Although most previous studies and meta-analyses have reported between 2.5- and 4-fold increased risks of GDM in women with PCOS, none of these studies were able to determine if this effect is independent of overweight/obesity, which is a major confounder [38,39,40]. As mentioned above, our findings suggest that the observed pregnancy complications in women with PCOS are more likely to be associated with the high BMI rather than PCOS. 

In contrast with our findings, a recent large database population study by Mills et al., published in 2020, reported that the observed 2-fold increase in GDM in women with PCOS was independent of obesity [41]. However, while our study included only IVF pregnancies, Mill and co-workers included all forms of pregnancies, with only 2.4% resulting from IVF. Furthermore, this study retrospectively utilised an administrative set of databases, which raises concerns regarding the accuracy and consistency of the data. 

On the other hand, in a recent comprehensive review published in 2023, Kotlyar and Seifer concluded that all women with PCOS undergoing IVF are at a higher risk of adverse pregnancy events such as gestational hypertension and that a high BMI can further exacerbate the overall risk [42]. The authors also reported that this exacerbating effect of high BMI can be eliminated by using frozen thaw embryo transfer (FET) for women with PCOS undergoing IVF, apart from the risk of caesarean delivery [42]. 

While obesity is known to elevate the risk of GDM due to IR [43,44,45], the precise relationship between PCOS and gestational hypertension remains more enigmatic. It is suggested that this relationship may be associated with the pathophysiological processes common to PCOS, which make women more likely to have metabolic syndrome. Notable characteristics of this syndrome are central obesity and increased IR [46,47,48]. Early studies evaluating the risks of gestational hypertension in women with PCOS showed a 3- to 4-fold increased risk [40,47]. However, these studies also discovered that women who developed gestational hypertension had a higher likelihood of developing GDM [38]. In addition, Lewandowska at el. found that an excessive pre-pregnancy weight was associated with a higher odds ratio for gestational hypertension and GDM compared to a normal BMI [49].

#### 4.1.3. Caesarean Section

Our results demonstrate a substantial elevation in the risk of caesarean section among individuals with PCOS who have a high pre-pregnancy BMI and undergo IVF. Boomsma et al. reported that this higher incidence of caesarean section is associated with obesity. Women with PCOS with a normal BMI have a caesarean section rate similar to that of age-matched controls [50]. Caesarean section is more frequently performed in women with GDM, possibly due to the increased incidence of foetal macrosomia ‘large for gestational age,’, which is possibly caused by the elevated maternal glucose levels [51].

### 4.2. Limitations and Strengths

The main limitation of our meta-analysis is the retrospective design of most of the included studies, which is known to introduce several sources of bias. Another limitation is the variations in definitions used for different pregnancy outcomes, which could increase heterogeneity between studies.

On the other hand, our systematic review has several strengths that lend validity to the results. Firstly, the studies included a relatively large number of women with PCOS undergoing IVF (*n* = 7680). Secondly, all studies apart from one [32] included in the meta-analysis adopted the Rotterdam criteria for the diagnosis of PCOS, thereby reducing heterogeneity between studies and enhancing generalizability.

Further well-designed large prospective cohort studies are necessary to further assess the impact of a high BMI in women with PCOS undergoing IVF on pregnancy outcomes and complications. It is also important for future research to investigate the intricate relationship between the weight status and IVF outcomes in women with PCOS. 

### 4.3. Interpretation of the Results

Several factors might explain the increased miscarriage rates in women with PCOS with a high BMI. Firstly, obesity tends to increase the risk of IR [43,44,45]. Proper glucose metabolism is crucial for endometrial decidualization, and IR could modify endometrial receptivity [52]. Secondly, this trend might be linked to persistent inflammatory states. Oróstica et al. identified consistent serum levels of tumour necrosis factor alpha (TNF-α) but amplified TNF-α signalling with NFκB in the endometrium of women with PCOS with a higher BMI [53]. Xue et al. offered a theoretical framework highlighting the abnormal endometrial conditions in women with PCOS to elucidate their subfertility issues. This includes issues caused by anovulation leading to endometrial hyperplasia, the impact of high androgen levels hindering endometrial development, differentiation, and decidualization, and disruptions in glucose metabolism in the endometrium due to insulin resistance, which in turn affects its receptivity. Additionally, they pointed out the endometrial resistance to progesterone and a persistent inflammatory state, both of which create a self-perpetuating cycle that alters the endometrial hormonal and metabolic environment, thereby hindering its receptivity [54]. Additionally, obesity is linked to compromised ovarian function, poor oocyte quality, and reduced reproductive outcomes. This is due to increased levels of proinflammatory cytokines like interleukin 6 (IL-6) and TNFα, along with oxidative stress [55,56]. Other researchers have proposed that the increased miscarriage rate this is linked to increased LH levels [57]

The reduced live birth rate in women with PCOS with a high BMI could be explained by the higher miscarriage rate in this group compared to the normal weight group (Figure 3). Being overweight or obese could also lead to a diminished live birth rate due to impaired decidualization, which in turn may cause abnormalities in implantation [58]. Other possible adverse effects of a high BMI that could compromise live birth rates include dyslipidaemia and hyperinsulinemia [59].

### 4.4. Clinical Implications

Women with PCOS with a high BMI undergoing IVF should be advised about the increased risks of adverse pregnancy outcomes. Clinicians should proactively assess the weight status of women with PCOS seeking fertility treatments, educate them about the potential impact of weight on IVF outcomes, and empower them to actively engage in lifestyle modifications. It is important to include weight management interventions as part of the care for women with PCOS planning to undergo IVF. On the other hand, women with PCOS with a normal BMI could be reassured about their future pregnancy outcomes, which are expected to be similar to those in women without PCOS. 

## 5. Conclusions

The findings of this meta-analysis suggest that the previously reported increase in pregnancy complications in women with PCOS undergoing IVF seems to be attributed to the PCOS-related high BMI rather than PCOS itself. Weight reduction is recommended before IVF to avoid possible potential adverse pregnancy complications.

### BMI and Multiple Pregnancy in PCOS Women Undergoing IVF

The pooled data from four studies (*n* = 1277) indicate no statistical difference between women with PCOS with a high BMI undergoing IVF compared with those with a normal BMI (OR 1.12 [95% CI, 0.88–1.43]; z = 0.95; *p* = 0.34; I^2^ = 0%). No heterogeneity was observed among the included studies.

## Figures and Tables

**Figure 1 jcm-13-01578-f001:**
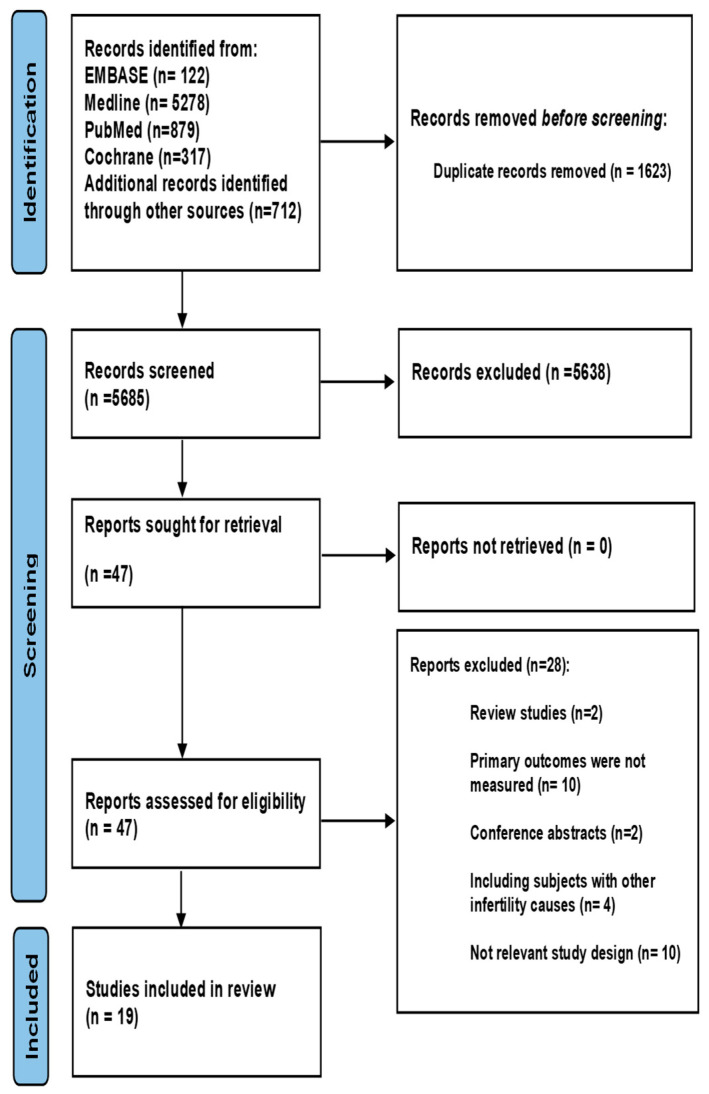
PRISMA flow chart.

**Figure 2 jcm-13-01578-f002:**
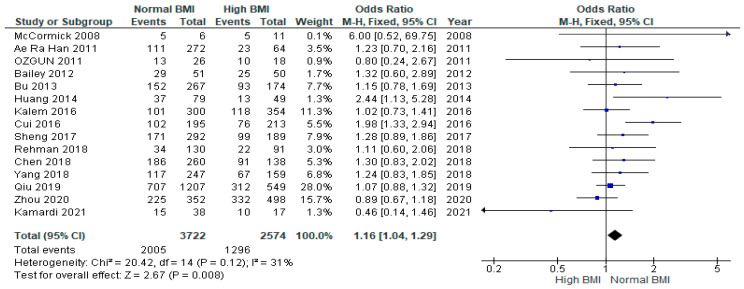
Clinical pregnancy rates pooled analysis of 15 studies [14,15,16,17,18,19,21,22,23,24,25,27,28,30,31].

**Figure 3 jcm-13-01578-f003:**
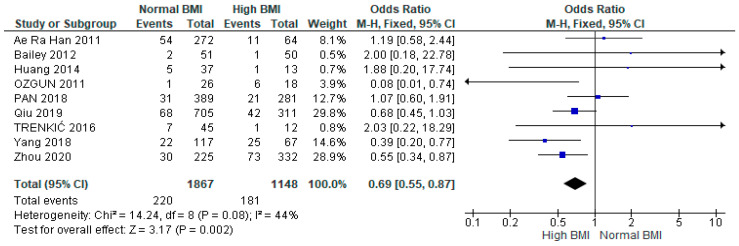
Miscarriage rates pooled analysis of data from nine studies [14,15,16,17,21,23,26,29,31].

**Figure 4 jcm-13-01578-f004:**
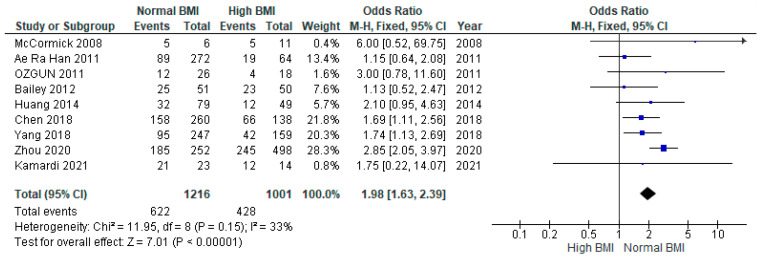
Live birth meta-analysis of nine studies [14,16,17,18,19,21,23,28,31].

**Figure 5 jcm-13-01578-f005:**
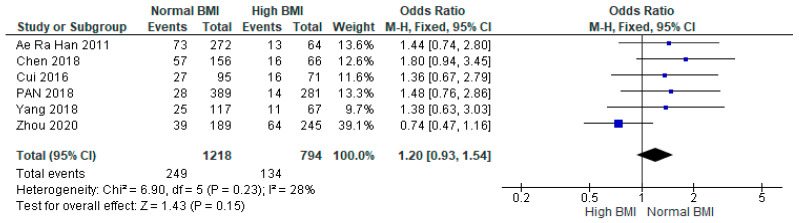
Preterm birth data analysis of 6 studies [14,16,24,28,29,31].

**Figure 6 jcm-13-01578-f006:**
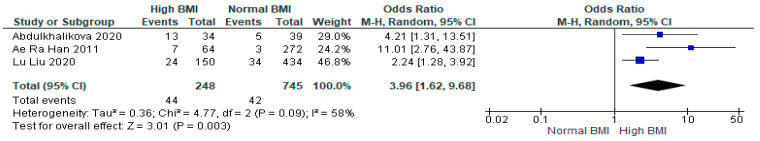
Gestational diabetes mellitus in women with PCOS undergoing IVF meta-analysis of data from three studies [16,32,33].

**Figure 7 jcm-13-01578-f007:**
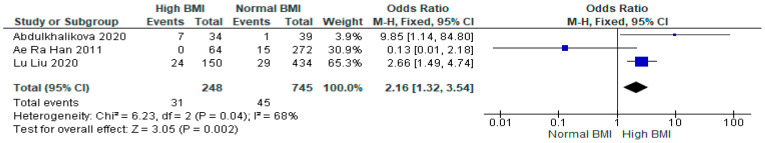
Gestational hypertension meta-analysis of three studies [16,32,33].

**Figure 8 jcm-13-01578-f008:**
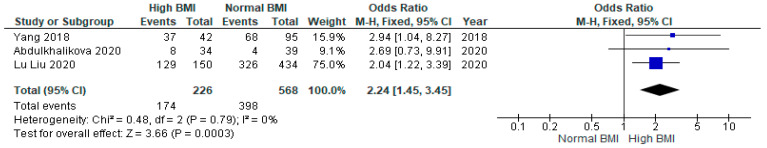
Caesarean section meta-analysis of three studies [31,32,33].

**Table 1 jcm-13-01578-t001:** Characteristics of the 19 studies included in the systematic review.

1st Author, Year	Country	Study Design	Study Population						Measured Outcomes	IVF Stimulation Protocol	NOS Score
			BMI (kg/m)		Sample Size		Age (y)				
			Normal	High	N-BMI	H-BMI	N-BMI	H-BMI			
McCormick, 2008 [18]	USA	RC	<25	≥30	6	11	31.5 ± 3.0	31.0 ± 5.0	CP/LB	Long agonist (98%); Antagonist (2%)	8
Ozgun, 2011 [21]	Turkey	PC	<30	≥30	26	18	26.8 ± 4.5	26.7 ± 2.9	CP/BP/M/LB	Long Agonist	8
Huang, 2014 [17]	China	RC	<24	≥24	79	49	30.5 ± 4.1	29.4 ± 3.4	CP/M/LB	ND	7
Bailey, 2014 [23]	USA	RC	<25	≥25	51	50	32 ± 3.5	32.4 ± 3.2	CP/BP/M/LB	ND	8
Cui, 2016 [24]	China	RC	<24	≥24	195	213	26.99 ± 3.3	27.5 ± 3.4	CP/BP/PTB	Long Agonist	6
Sheng, 2017 [27]	China	PC	<24	≥24	292	189	28.5 ± 2.7	26.3 ± 3.1	CP	Long Agonist	7
Pan, 2018 [29]	China	RC	<24	≥24	389	281	27.7 ± 3.17	29.4 ± 3.53	M/PTB	Long Agonist	6
Yang, 2018 [31]	China	RC	<25	≥25	247	159	29.1	29.5	CB/LB/PTB/CS	Antagonist	7
Chen, 2018 [28]	China	RC	<24	≥24	260	138	28.8 ± 2.7	28.9 ± 3.0	CP/LB/PTB	Antagonist	7
Qiu, 2019 [15]	China	RC	<25	≥25	1207	549	28.91 ± 3.2	30.0 ± 3.6	CP/M	Antagonist	6
Zhou, 2020 [14]	China	RC	<25	≥25	352	498	27.7 ± 2.5	27.9 ± 3.1	CP/MP/M/LB/PTB	Ultra-Long Agonist	7
Rehman, 2018 [30]	Pakistan	CS	<25	≥25	130	91	32.4 ± 4.4	32.0 ± 4.8	CP	Long protocol	8
Kalem, 2016 [25]	Turkey	RC	<25	≥25	300	354	ND	*	CP/MP	Long agonist/Antagonist	7
Bu, 2013 [22]	China	RC	<24	≥24	267	174	28.3 ± 4.0	29.1 ± 4.1	CP	Long Agonist	7
Ae Ra Han, 2011 [16]	Korea	RC	<25	≥25	272	64	31.2 ± 2.7	31.6 ± 3.1	CP/MP/M/LB/PTB/GDM/GH	Long Agonist	8
Kamardi, 2021 [19]	Indonesia	RC	<25	≥25	38	17	30.1 ± 4.0	30.4 ± 4.3	CP/BP	Long Agonist	8
Abdulkhalikova, 2020 [33]	Slovenia	RC	<25	≥25	39	34	33.4 ± 4.2	33.5 ± 4.0	GDM/GH/CS	ND	6
Liu, 2020 [32]	China	RC	<24	≥24	434	150	29 (27–32)	31 (27–34)	GDM/GH/CS	ND	7
Trenkić, 2016 [26]	Serbia	RC	≤25	>25	45	12	31.6 ± 3.99	31.5 ± 4.3	M/GDM/GH	Long agonist (62%); Flexible agonist (38%)	8

Data are presented as mean ± sd; (range). * Age < 35 for the overweight group (*n* = 208) and ≥35 for the obese group (*n* = 146); RC, retrospective cohort; PC, prospective cohort; CP, clinical pregnancy; BP, biochemical pregnancy; MP, multiple pregnancy; M, miscarriage; LB, live birth; PTB, preterm birth; GDM, gestational diabetes millets; GH, gestational hypertension; CS, caesarean section; GnRH-anta, gonadotropin releasing hormone antagonist; ND, not documented.

## Data Availability

Not applicable.

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
