# Peer review of "The Impact of High BMI on Pregnancy Outcomes and Complications in Women with PCOS Undergoing IVF—A Systematic Review and Meta-Analysis"

_jcm, 2024, doi:10.3390/jcm13061578_

Round 1

Reviewer 1 Report

Comments and Suggestions for Authors

I read with great interest the present Manuscript which falls within the aim of the Journal. In my honest opinion, the topic is interesting enough to attract the readers’ attention. Methodology is accurate and conclusions are supported by the data analysis. Nevertheless, authors should clarify some points and improve the discussion citing relevant and novel key articles about the topic. For all those reasons, I suggested performing the minor revisions.

-    Exclusionary criteria should be clarified in the Methods section;

-    In Discussion, Authors may discuss their conclusions in the context of other evidence about the topic. Please consider: “Kotlyar AM, Seifer DB. Women with PCOS who undergo IVF: a comprehensive review of therapeutic strategies for successful outcomes. Reprod Biol Endocrinol. 2023;21(1):70. Published 2023 Aug 1. doi:10.1186/s12958-023-01120-7”;

-    The Authors have not adequately highlighted the strength of the study. I suggest better specifying that point.

Author Response

Comment: I read with great interest the present Manuscript which falls within the aim of the Journal. In my honest opinion, the topic is interesting enough to attract the readers’ attention. Methodology is accurate and conclusions are supported by the data analysis. Nevertheless, authors should clarify some points and improve the discussion citing relevant and novel key articles about the topic. For all those reasons, I suggested performing the minor revisions.

Response: we would like to thank the reviewer for the positive and encouraging comment.

Comment: Exclusionary criteria should be clarified in the Methods section

Response: we have already clarified the exclusion criteria under the methods section [lines 91-93].

Comment: In Discussion, Authors may discuss their conclusions in the context of other evidence about the topic. Please consider: “Kotlyar AM, Seifer DB. Women with PCOS who undergo IVF: a comprehensive review of therapeutic strategies for successful outcomes. Reprod Biol Endocrinol. 2023;21(1):70. Published 2023 Aug 1. doi:10.1186/s12958-023-01120-7”;

Response: we would like to thank the reviewer for this valid point, and we have added a statement referring to this reference [lines 309-314]. We have also updated the references in the text and in the list (as highlighted).

Comment: The Authors have not adequately highlighted the strength of the study. I suggest better specifying that point.

Response: we have expanded on the limitations and strengths of the study as suggested (lines 336 – 343).

Reviewer 2 Report

Comments and Suggestions for Authors

Dear, authors after reading your manuscript, I have some suggestions/questions:

1. Do not use abbreviations in the title

2. Explain abbreviation in Abstract

3. Row 42 - Provide percent from the reference

4. Row 103 - "up to June 2023..." since when?

5. Search strategy.... place the keyword between inverted commas. Were they MeSH (Medical Subject Headings)? Please specify

6. I would suggest to detail the NOS score in a different table

7. I think you have misinterpreted the heterogeneity in some cases. If you refer to the Cochrane handbook (https://handbook-5-1.cochrane.org/chapter_9/9_5_2_identifying_and_measuring_heterogeneity.htm), 28% (Figure 5) It is not moderate, it is actually low

8. You should highlight the strengths and limitations of the study

Author Response

Comment: Do not use abbreviations in the title

Response: thanks for this valid comment. However, removing the abbreviations will make the title so long (>3 lines). All the abbreviations in the title are standards and widely used in the literature. Furthermore, these standard abbreviations are now increasingly used in the titles in the literature as shown in the following examples:

Kotlyar and Seifer, 2023 - Reprod Biol Endocrinol. https://doi.org/10.1186/s12958-023-01120-7). 

Dobbie et al, 2023 (Human Reproduction - https://doi.org/10.1093/humrep/dead053)

Wei et al 2022 (Human Reproduction - https://doi.org/10.1093/humrep/deac154)

Moolhuijsen et al 2022 (Human Reproduction - https://doi.org/10.1093/humrep/deac082)

It will therefore be better if the title stays as it is.

Comment: Explain abbreviation in Abstract

Response:  all done

Comment: Row 42 - Provide percent from the reference

Response:  percent added based on the reference

Comment: Row 103 - "up to June 2023..." since when?

Response:  No date earlier date limit applied

Comment: Search strategy.... place the keyword between inverted commas. Were they MeSH (Medical Subject Headings)? Please specify

Response:  Commas have been used instead of brackets - yes the terms were MeSH and this has been specified.

Comment: I would suggest to detail the NOS score in a different table

Response: a new table has been added to the supplementary document (Table S1).

Comment: I think you have misinterpreted the heterogeneity in some cases. If you refer to the Cochrane handbook (https://handbook-5-1.cochrane.org/chapter_9/9_5_2_identifying_and_measuring_heterogeneity.htm), 28% (Figure 5) It is not moderate, it is actually low.

Response: corrected, thank you for this observation.

Comment: You should highlight the strengths and limitations of the study.

Response: we have expanded on the limitations and strengths of the study as suggested (lines 336 – 343).